# CAUSALBENCH CHALLENGE: DIFFERENCES IN MEAN EXPRESSION

**Marcin Kowiel, Wojciech Kotlowski & Dariusz Brzezinski**
Institute of Computing Science
Poznan University of Technology
`{dbrzezinski,wkotlowski}@cs.put.poznan.pl`

## ABSTRACT

In this write-up, we describe our solution to the 2023 CausalBench Challenge. We describe our approaches to preprocessing the data, parameterizations of DCDI and GRNBoost, and modifications to the baseline algorithms.

## 1 DATA PRE-PROCESSING AND POST-PROCESSING

Parallel to developing modifications of the baseline DCDI and GRNBoost algorithms, we considered modifications to the input and output data of these algorithms. In particular, we analyzed good initial values for the gene expression threshold and output graph size.

**Gene expression threshold.** The gene expression threshold is used to remove genes that have a non-zero expression in less than a user-defined fraction of the samples. The default value of 0.25 resulted in DCDI performance that was visibly worse than that reported in (Chevalley et al., 2022). Therefore, we changed the default expression threshold to 0.5 and used this value in further experiments. Moreover, we omitted samples labeled as 'excluded'.

**Output graph size.** The challenge submissions are evaluated based on the mean Wasserstein distance between the expression distributions of connected pairs of nodes in the output graph. Seeing that not all pairs are equally important and methods such as GRNBoost rely on sorting pairs according to importance and then selecting only a subset of them using a threshold, we assumed that smaller graphs will be more likely to have a higher value of the mean Wasserstein distance. To verify this hypothesis, we plotted the mean Wasserstein distance for GRNBoost graphs of different sizes. As can be noticed by looking at Figure 1, the mean Wasserstein distance indeed decreases as the number of edges in the graph grows. Although GRNBoost does not perfectly sort gene pairs according to differences between expression distributions, the results are still very good.

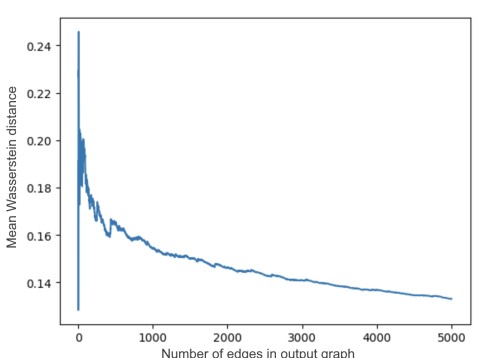

Figure 1: Mean Wasserstein distance for different sizes of GRNBoost output graphs.

Therefore, in further experiments, we have always limited the number of edges to 1,000, which was the smallest allowed output graph according to the competition rules.

## 2 TESTED APPROACHES

In this section, we will discuss the subsequent models we tested while preparing our final Causal-Bench Challenge submission.

**DCDI and GRNBoost baselines.** Before designing modifications, we ran experiments on DCDI (Brouillard et al., 2020) and GRNBoost (Huynh-Thu et al., 2010) with different parameters. As mentioned in the previous section, we finally settled for a gene expression threshold of 0.5 and an output graph consisting of 1,000 edges. We also tested different versions of DCDI-G (which offered better performance than DCDI-DSF). As can be seen in the left panel of Figure 4, our results on the RPE dataset (Tsherniak et al., 2017) are in accordance with those presented in (Chevalley et al., 2022), i.e., DCDI-G offers the best performance followed by GRNBoost. These results served as a reference point for our modifications.

**GRNBoost with intervention encoding.** Our first modification involved adding information about interventions to GRNBoost. GRNBoost creates multiple regressors, each one predicting the expression value of a gene based on the expression values of the other genes. In its original form, GRNBoost treats all samples equally and has no notion of gene interventions. The first and simplest modification involved changing the expression value of perturbed genes to -100 (Figure 2, left panel). By doing so, our goal was to differentiate between interventions and naturally occurring zero-expression of a given gene. Since GRNBoost relies on regression trees, we did not worry about the concrete intervention encoding value, as long as it separated interventions from observational values. Hence we only tested the value -100. The experimental results of this modification for the RPE dataset are presented in the right panel of Figure 4. As can be noticed, the intervention encoding strategy offered slightly better performance than the baseline GRNBoost.

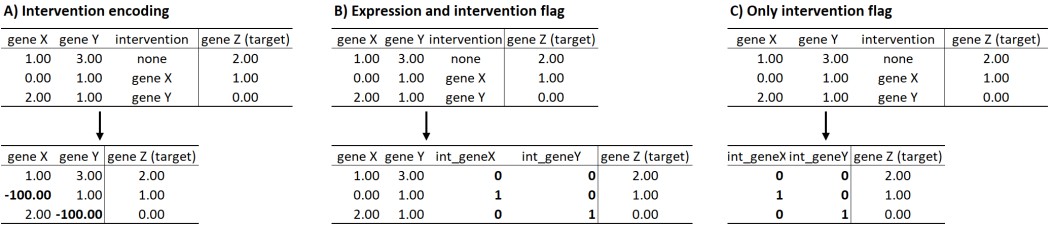

Figure 2: Schematic of data modifications performed to introduce intervention information to GRNBoost. Each table presents the dataset used to train one regressor to predict the expression of gene Z based on expression values of genes X and Y.

**GRNBoost with intervention flag columns.** The approach described in the previous paragraph has a downside in that the intervention encoding value -100 hides the true expression of the gene in the sample, thus removing some of the information from the dataset. Therefore, as our second modification, instead of replacing expression values, we have added a set of columns with binary flags determining whether a particular gene was perturbed in a given sample (Figure 2, center panel). Somewhat surprisingly, this strategy of extending the dataset performed worse than intervention encoding (Figure 4).

**GRNBoost with only intervention flag columns.** Since extending the dataset with more columns seemed to have added more noise, we also tried another strategy—one wherein we discarded the expression values altogether and left only intervention flags (Figure 2, right panel). This GRNBoost modification worked significantly better than the previous two (Figure 4). Since using only binary (one-hot) intervention flags to predict expression boils down to estimating the means for sub-populations of the dataset, we decide to test strategies that estimate mean expression directly.

**Mean expression estimation.** We measured the strength of causal relationship $X \rightarrow Y$ for every gene pair $X, Y$, for which interventions on $X$ were available. To this end, we separately calculated for gene $Y$ its mean expression values $\bar{Y}_O$ and $\bar{Y}_X$ on the observational data and on the interventional data concerning perturbations of $X$, respectively. The difference in means, $|\bar{Y}_O - \bar{Y}_X|$, was used to measure the strength of the relationship, and then to sort gene pairs and select 1,000 pairs with the largest differences. It turns out that this simple approach, which is essentially a regression model of $Y$ on the intervention flag of $X$, turned out to significantly outperform all previously tested strategies on the RPE dataset, as seen in Figure 3. We note that for mean difference estimation, we did not employ any gene expression threshold.

**Mean expression estimation with Bayesian correction.** Since some of the interventions contained few samples, we decided to correct the mean expression value on the interventional data $\bar{Y}_X$ by employing a Bayesian estimator, treating $\bar{Y}_O$ as the prior mean, and the variance of $Y$ on the observational data, $\text{Var}(Y_O)$, as the prior variance. This effectively boils down to expressing the difference in means by $c_{XY}|\bar{Y}_O - \bar{Y}_X|$, with Bayesian correction factor $c_{XY} = \frac{\text{Var}(Y_O)}{\text{Var}(Y_O) + \text{Var}(Y_X)/n_X}$, where $\text{Var}(Y_X)$ is the variance of $Y$ on the interventional data concerning perturbations of $X$, while $n_X$ is the number of samples in that intervention. Since $c_{XY} \leq 1$ and increasing with $n_X$, this has the effect of discounting the mean differences for

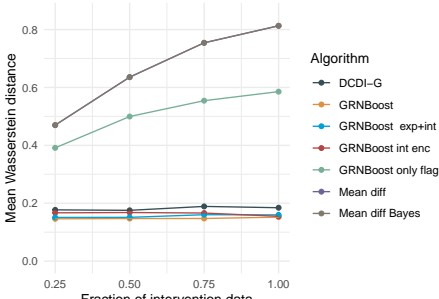

Figure 3: Comparison of all algorithms. Note that both mean difference methods (Mean diff, Mean diff Bayes) have practically the same performance.

small interventional datasets. However, the Bayesian estimation brought only an insignificant improvement when compared with the previous approach, essentially returning an almost identical set of top 1,000 pairs (Figure 3).

Considering all of the experimental results (Supplementary Table 1) and the above analyses, our final submission consisted of omitting samples labeled as 'excluded', estimating the mean expression of genes for each intervention, and selecting the 1,000 gene pairs with the largest expression differences.

## 3 Discussion

Our final submission consisted of a very simple algorithm that estimates the mean expression of genes in situations when a different gene is intervened upon. The reason why we have settled for such a simple method rather than a more elaborate one stems from three factors:

1. the fact that this is a competition, not an exploratory analysis;
2. the format of the training and testing data;
3. the competition's evaluation metric.

The first factor is obvious: since we are participating in a competition, discovering new interesting causal gene relationships becomes less important than achieving the best performance according to the competition rules. During the explanatory analysis and tests of various approaches, we realized that every step which led to the performance improvements was essentially pulling a given method towards estimating the difference in means on the observational and the interventional data. Therefore, we eventually decided to use the mean estimation as the sole method for causal graph edge prediction. The second factor, the data format, required us to predict interactions only between genes that were present in the input data and which, in most cases, had interventions. Without expression data for genes without interventions, there was no reason to predict causality between unperturbed genes. Finally, for a well-behaved predictor, the value of the competition evaluation metric will decrease as the number of predicted edges increases; therefore, it was always optimal to predict as few gene interactions as the competition allowed.

The above factors made our submission much simpler, but also much less applicable to industry needs. To alleviate the above-mentioned issues, we believe it would be necessary to require contestants to predict causal relations between genes that have interventions as well as those pairs that are purely observational. For that to be possible, the input data should have more genes without any observations, and the algorithm should receive as input the pairs of genes it is going to be evaluated on. With such a setup, the organizers would be able to force predictions on observational genes from the training data and evaluate them based on held-out interventional data. By prespecifying, which gene pairs the algorithm is supposed to assess, the problem of predicting the smallest possible graph would also disappear. In general, gene pairs could be evaluated in three cross-validation or holdout settings (`cv1`, `cv2`, `cv3`) as proposed for synthetic lethality pairs by Wang et al. (2022).

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

## A  APPENDIX

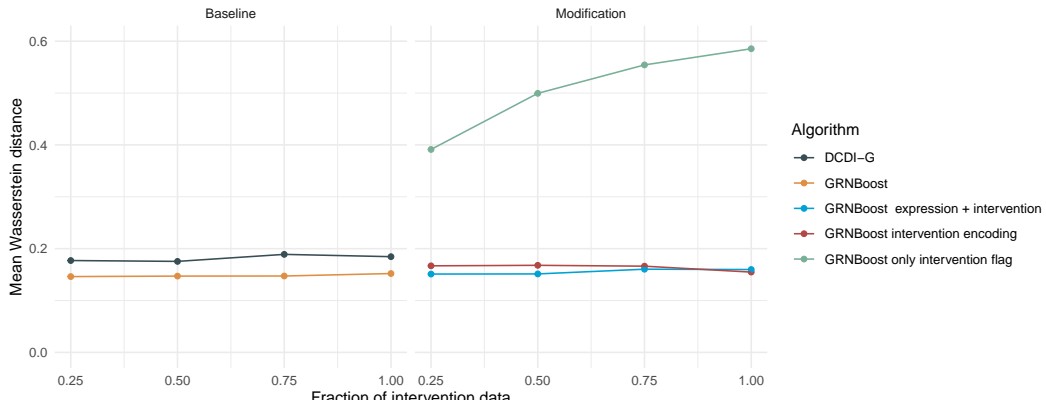

Figure 4: Mean Wasserstein distance of baseline algorithms and modifications of GRNBoost on the RPE dataset. Left panel: baseline algorithms—DCDI-G and GRNBoost. Right panel: GRNBoost modifications.

Table 1: Experimental results of all the algorithms on the RPE dataset.

| Algorithm | Fraction of intevention data | Mean Wasserstein distance |
|---|---|---|
| DCDI-G | 0.25 | 0.1771 |
| DCDI-G | 0.50 | 0.1755 |
| DCDI-G | 0.75 | 0.1890 |
| DCDI-G | 1.00 | 0.1845 |
| GRNBoost | 0.25 | 0.1462 |
| GRNBoost | 0.50 | 0.1471 |
| GRNBoost | 0.75 | 0.1473 |
| GRNBoost | 1.00 | 0.1520 |
| GRNBoost intervention encoding | 0.25 | 0.1669 |
| GRNBoost intervention encoding | 0.50 | 0.1679 |
| GRNBoost intervention encoding | 0.75 | 0.1662 |
| GRNBoost intervention encoding | 1.00 | 0.1548 |
| GRNBoost expression + intervention | 0.25 | 0.1510 |
| GRNBoost expression + intervention | 0.50 | 0.1513 |
| GRNBoost expression + intervention | 0.75 | 0.1604 |
| GRNBoost expression + intervention | 1.00 | 0.1598 |
| GRNBoost only intervention flag | 0.25 | 0.3913 |
| GRNBoost only intervention flag | 0.50 | 0.4995 |
| GRNBoost only intervention flag | 0.75 | 0.5542 |
| GRNBoost only intervention flag | 1.00 | 0.5855 |
| Mean diff | 0.25 | 0.4697 |
| Mean diff | 0.50 | 0.6357 |
| Mean diff | 0.75 | 0.7541 |
| Mean diff | 1.00 | 0.8130 |
| Mean diff Bayes | 0.25 | 0.4699 |
| Mean diff Bayes | 0.50 | 0.6354 |
| Mean diff Bayes | 0.75 | 0.7542 |
| Mean diff Bayes | 1.00 | 0.8128 |

