# OpenReview forum: "CausalBench Challenge: Differences in Mean Expression"
_GSK.ai/2023/CBC_

### Official Review · Reviewer_brEv · 2023-04-25

**Rating:** 7
**Confidence:** 5

**Review:**

The team thoroughly reported on their different attempts and discoveries around the task, as well as how to optimise the evaluation metrics. They proposed different approaches to adding the interventional information to GRNBoost. In the end, they noted that directly optimising for the evaluation metric through Mean Difference estimation lead to the best score. Unfortunately, as reported also by the team, this algorithm do not try to uncover the causal graph among genes, as only edges with an intervened gene as parent can be predicted. Also, predicted edges are not guaranteed to be direct edges. This method is similar to the Difference in differences method, which is used to estimate causal effect, but not causal structure.

The team thus provides a thorough analysis of the evaluation metrics weaknesses and also made interesting propositions as to how to improve it. Unfortunately, as was explicitly written in the challenge rules, exploiting such knowledge for the method was strictly forbidden. As such, we have decided to only consider the "GRNBoost expression + intervention" submission for the final ranking. We thank the team for their understanding as well as for the effort put into the challenge.